# Reproducibility of Contrastive Clustering

**Abstract**

In the paper *"Contrastive Clustering"* (AAAI 2021) [1], authors proposed an innovative contrastive learning algorithm in clustering, namely Contrastive Clustering (CC), which can remarkably outperform 17 traditional clustering methods including K-means, PICA in all evaluation metrics. Unlike traditional clustering, CC simultaneously conducts the instance-level and cluster-level contrastive learning in the row and column space of feature matrix, leading to an improvement of clustering performance. In order to verify the performance of their CC algorithm, we reproduce results from their paper. Specifically, we re-implement the results of Figure 3, Table 2, and Figure 4 from the paper to evaluate the reproducibility. Then We apply CC on other datasets to test the robustness of CC algorithm. In addition, we conduct ablation study to examine the importance of contrastive heads to the model performance. Finally, we successfully improve the performance of proposed CC model by modifying their loss function.

## 1 Introduction

As illustrated in paper *"Contrastive Clustering"* (AAAI 2021), the existing studies have only performed contrastive learning at the instance level, whereas the proposed Contrastive Clustering (CC) conducts contrastive learning at both the instance-level and the cluster-level. Such a dual contrastive learning framework could improve the clustering dramatically. To prove its performance, we conduct our project from four angles: Reproducibility, Applying the model to other datasets, Ablation study, and Improving the performance of the proposed method.

For reproducibility part, we will reproduce their experiments and results: pair-wise similarity analysis (paper's Figure 3), Clustering performance comparison (paper's Table 2), and evolution of training process (paper's Figure 4). Due to the GPU and time limitation, we conduct some experiments with smaller epoch and batch size. For Ablation study, We selectively choose one of their ablation studies to prove the importance of cluster-head to the model. In addition to their study, we apply their CC model on different datasets, including CIFAR-10, CIFAR-100, ImageNet-10, to measure the robustness of the model. Last but not least, we improve their proposed model by changing the loss function and obtain an even better result.

## 2 Paper Summary

The framework of Contrastive Clustering can be presented in Figure 1. CC consists of three jointly learned components, namely a pair construction backbone (PCB), an instance-level contrastive head (ICH), and a cluster-level contrastive head (CCH).At beginning, PCB will construct positive and negative data pairs using two data augmentations (ResizedCrop, ColorJitter, GaussianBlur, etc.). Then a shared deep neural network is used to extract features from 2 different augmentations. Thereafter, two separate MLPs ($\sigma$ denotes the ReLU activation and $\sim$ denotes the Softmax operation to produce soft labels) are used to project the features into the row and column space wherein the instance- and cluster-level contrastive learning are conducted respectively. The main objective of contrastive learning here is to maximize the similarity between those positive pairs and minimize the similarity between those negative ones. This contrastive learning objective can be achieved on both instance-level and cluster-level by simply minimizing the addition of their contrastive loss function (Eq.1).

After constructing the CC model, authors attempt to evaluate its performance in terms of three metrics: Normalized Mutual Information (NMI), Accuracy (ACC), and Adjusted Rand Index (ARI). After comparing the accuracy of the model to 17 other clustering approaches, the paper concludes that the CC approach outperforms all of them remarkably. They also change the dataset and repeat this experiment, and they eventually get a similar result, which can prove CC has a high robustness.

Finally, the authors conduct ablation experiments to test the importance of data augmentation, the effect of two contrastive heads and the reliance on the backbone network. They conclude that the performance of

contrastive learning relies heavily on the choice of data augmentation, the joint effects of ICH and CCH is the key reason why CC can outperforms other clustering methods, and a deeper network does not promise a better performance.

# 3 Our Experiment Results

## 3.1 Reproducibility

### 3.1.1 Pair-wise similarity analysis

We achieve the pair-wise similarity analysis result as Figure 2 (Appendix). This is the reproduction of the Figure 3 of the paper. In general, our reproduced result shows that the similarity of positive instances and clusters increases over the training process while similarity of negative instances and clusters stays at low level and converges after 50 epochs. This could be the evidence that the contrastive learning objective is achieved because the similarities of positive pairs are being maximized while those of negative pairs are being minimized throughout the training. Thus, we conclude our result matches the paper's result at this point, and the CC does possess contrastive learning ability on clustering.

There is a small difference between our figure and paper's figure for the negative cluster instance pairs. In paper's figure, similarity of negative samples first increases then decreases while our similarity of samples directly plunge to 0, which suggests our result provides a better evidence of "minimizing negative pairs" objective.

### 3.1.2 Clustering performance comparison

Table 1 (Appendix) shows high resemblance between our reproduced table and paper's table2. Due to GPU and time limitation, we only select K-means and CC methods, CIFAR-10 and CIFAR-100 datasets to conduct the performance investigation. We used three evaluation metrics (NMI, ACC, ARI) to compare the CC to other clustering methods. As shown in the our table, the performance of our K-means method is approximately equal to the paper's result while that of CC method from our result is exactly the same as paper's result. This is because we used the model given by the author, to get the performance of CC while K-means method is created on our own. Thus we can conclude that their table highly matches with what we reproduced, and it's safe to say their table has a high credibility.

Most importantly, based on their table and our reproduced table, there's a huge difference between the accuracy of CC method and K-means method. That verifies the statement that "CC remarkably outperforms traditional clustering method". CC takes the advantage of both instance-level and clustering-level contrastive learning to obtain a much better result.

### 3.1.3 Evolution of Instance Feature and Cluster Assignments

In this section, we conduct a reproducing experiment to check that if the CC model can gradually differentiate different groups over training process as what paper's Figure4 suggest. And we obtain the reproduced results as Figure 3, Figure 4, and Figure 5 (Appendix). We use ImageNet-10 dataset to reproduce the result. Note that data points must be gone through backbone network first before using for plotting with t-SNE.

Although our reproducing result is not totally the same with the result from the paper, in general, we can see that image objects are clustered clearly when the number of epoch increases. In case 0 epoch, boundary area of clusters indicated by different colors is very close and could be partially overlapped. When the number of epoch increase to 20, different objects are clustered clearer. The objects are separated more distinctly when increasing the number of epoch increase to 100. In the paper, they trained the model using 1000 epochs, so the clustering result can only be better at that level.

## 3.2 Applying the model to other datasets

In order to testify the effectiveness of contrastive learning in CC model, we apply the pair-wise similarity analysis not only on ImageNet-10 like what authors did, but also on CIFAR-10 dataset to see if the contrastive learning universally applied.

The result is presented in Figure 6 (Appendix). Just like Figure 2 (Appendix), the positive pairs are being maximized and negative pairs are being minimized. And we can see on CIFAR-10 dataset, the contrastive learning is more stable and robust than ImageNet, the similarity of positive pairs constantly stays at high level while the similarity of negative pairs constantly stays at low level. That's because the CIFAR-10 only has 10 categories while ImageNet has much more groups and it will be more uncertain for CC algorithm at beginning. Thus we can deduce that the success of contrastive learning on ImageNet will be also applied to other datasets including CIFAR-10. At this point, we have proved the robustness of CC method.

## 3.3   Ablation Study

The paper conducted three ablation Studies. We select one of them to reproduce. We carry out a reproducing experiment to check the effect of contrastive heads to model performance. The reason for us to specifically choose this one is that the dual joint effect of two contrastive heads is the unique part which differentiates CC model from traditional contrastive learning, thus we want to verify how much does contrastive heads contribute to the performance. This experiment is corresponding to section 4.4.2 of the paper. To save time, we choose ImageNet-10 dataset and set the number of epoch to 20 for the training process. Table 2 (Appendix) show that although our reproduced performance is significantly different from their original table, the trend we reproduced is exactly the same as theirs. The difference between them is because our training iterations use 20 epoch while their model was trained with 1000 epoch.

Based on the same pattern both tables present, we can easily see that combination of two contrastive heads in a model gives better performance than using them individually. More importantly, using cluster contrastive head individually performs much better than using instance head individually, which suggest the cluster head plays a more important role in increasing the CC's performance. This could explain why CC outperforms remarkably than traditional contrastive learning as well, because none of existing study has applied cluster contrasitive learning into their model. Therefore, the joint effects of ICH and CCH is the key to the success of CC algorithm.

## 3.4   Improving the performance of the proposed method

We get inspired by the previous ablation study, where authors tried to adjust the loss function to prove the importance of two contrastive heads. But in their original CC model, they reflected the joint effect of two contrastive heads by simply adding instance loss and cluster loss together. After investigating, we decide to improve the performance of CC method by modifying loss function. In this way, we hope to reflect the joint effect in a better mathematical relationship. Currently, the paper design a loss function equal to sum of loss components.

$$\mathcal{L} = \mathcal{L}_{ins} + \mathcal{L}_{clu} \tag{1}$$

where $\mathcal{L}_{ins}$ and $\mathcal{L}_{clu}$ are loss function of instance-level head and cluster-level head respectively. We try there approaches for loss function as follow.

$$\mathcal{L}_{avg} = \mathbf{Average}(\mathcal{L}_{ins}, \mathcal{L}_{clu}) \tag{2}$$

$$\mathcal{L}_{max} = \mathbf{Max}(\mathcal{L}_{ins}, \mathcal{L}_{clu}) \tag{3}$$

$$\mathcal{L}_{min} = \mathbf{Min}(\mathcal{L}_{ins}, \mathcal{L}_{clu}) \tag{4}$$

We run the each new model with 20 epochs to make a horizontal comparison and obtain the Table 3 (Appendix). Based on our table, we can see that $\mathcal{L}_{min}$ and $\mathcal{L}_{max}$ do not improve the model performance, it even decreases the value of NMI, ACC, and ARI. By contrast, $\mathcal{L}_{avg}$ makes a significant improvement, which increases NMI, ACC, and ARI of proposed approach to amount of 24,71%, 10,6%, and 31,33% respectively. In conclusion, $\mathcal{L}_{avg}$ is an outstanding solution to improve the proposed model.

Since we only use 20 epochs to train model with Eq.1 and model with Eq.2, we decided to run 100 epochs on each one to prove our model will outperform the proposed one regardless of the epochs. And it turns out the accuracy of proposed model is 0.801, and that of our improved model is 0.816. Therefore, our new model obtains a better performance than the proposed model.

# 4   Conclusion

In this project, we investigate the paper of Contrastive Clustering. We conduct reproducing experiments for pair-wise similarity analysis, clustering performance comparison, and evolution of training process, to validate the results for paper's Figure 3, Table 2, and Figure 4. Despite of the nuance due to the reduction in epochs and batch size, we obtain exctly the same results and pattern as they did. After that we also conduct ablation study to show that the combination of two contrastive heads in a model giving a better performance, and prove how the joint effect of two heads contribute to remarkable improvement to traditional methods. Last but not least, we improve the proposed model performance to a new level, which we demonstrate that loss function of proposed model should be calculated based on average loss to reflect the joint effect of contrastive learning, thus leads to an improvement of model performance.

# 5   Statement of Contributions

In this project, the workload can be broken down as: **Van Hau Le** and **Jiangshan Yu** conduct the reproducing experiments, ablation study and improve model performance. Haluk Calin investigate the paper content, collect information from experiment results and complete the report.

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

# Appendix

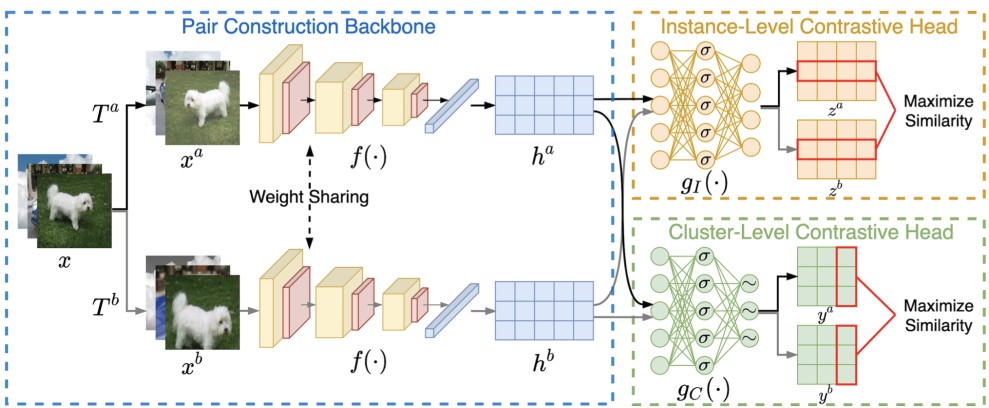

Figure 1: CC framework

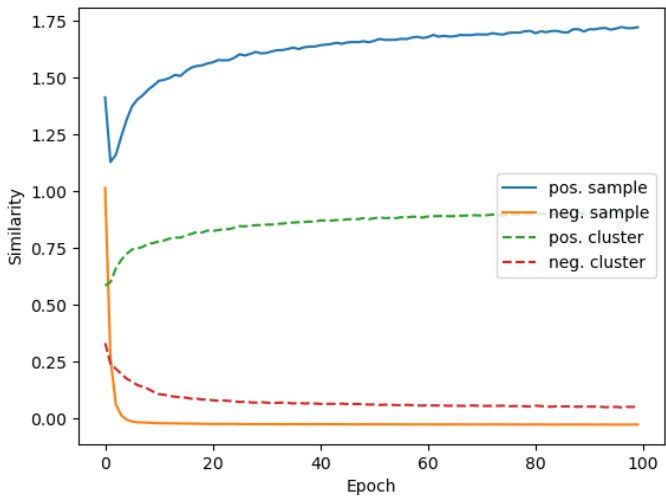

Figure 2: Pair-wise similarity analysis on ImageNet-10

| Our reproducing result | | | | | | |
|---|---|---|---|---|---|---|
| Dataset | CIFAR-10 | | | CIFAR-100 | | |
| Metrics | NMI | ACC | ARI | NMI | ACC | ARI |
| K-means | *0.082* | *0.232* | *0.051* | *0.076* | *0.131* | *0.029* |
| CC(ours) | *0.705* | *0.790* | *0.637* | *0.431* | *0.429* | *0.266* |
| | | | | | | |
| Result from paper's Table 2 | | | | | | |
| Dataset | CIFAR-10 | | | CIFAR-100 | | |
| Metrics | NMI | ACC | ARI | NMI | ACC | ARI |
| K-means | *0.087* | *0.229* | *0.049* | *0.084* | *0.130* | *0.028* |
| CC(ours) | *0.705* | *0.790* | *0.637* | *0.431* | *0.429* | *0.266* |

Table 1: Clustering performance comparison

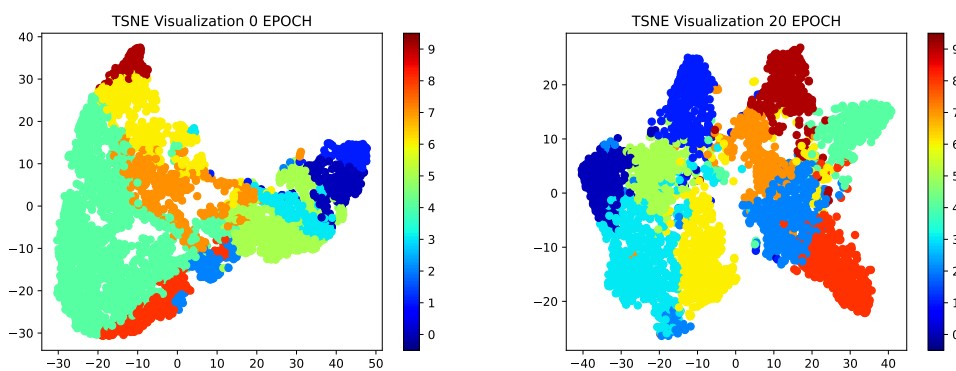

Figure 3: 0 EPOCH.                    Figure 4: 20 EPOCH

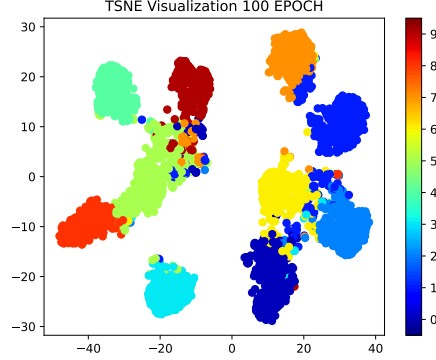

Figure 5: 100 EPOCH

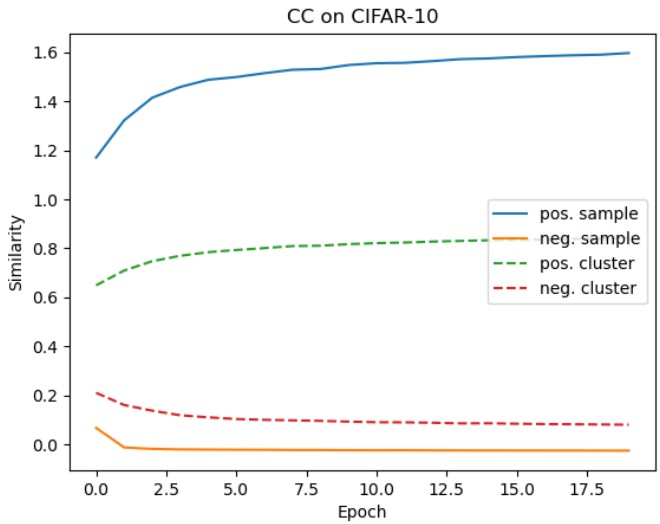

Figure 6: Pair-wise similarity analysis on CIFAR-10

| Our reproduced result | | | | |
|---|---|---|---|---|
| **Dataset** | **Contrastive Head** | **NMI** | **ACC** | **ARI** |
| **ImageNet-10** | **ICH + CCH** | *0.348* | *0.480* | *0.249* |
| | **ICH Only** | *0.140* | *0.221* | *0.061* |
| | **CCH Only** | *0.310* | *0.387* | *0.197* |
| | | | | |
| **Result from paper's Table 4** | | | | |
| **Dataset** | **Contrastive Head** | **NMI** | **ACC** | **ARI** |
| **ImageNet-10** | **ICH + CCH** | *0.859* | *0.893* | *0.822* |
| | **ICH Only** | *0.838* | *0.888* | *0.780* |
| | **CCH Only** | *0.850* | *0.892* | *0.816* |

Table 2: The effect of two contrastive heads

| Exploring different loss functions beyond proposed method | | | | |
|---|---|---|---|---|
| **Dataset** | **Loss Function** | **NMI** | **ACC** | **ARI** |
| **ImageNet-10** | **L = Instance-loss + cluster-loss (proposed)** | *0.348* | *0.480* | *0.249* |
| | **L = Average (Instance-loss, cluster-loss)** | *0.434* | *0.586* | *0.327* |
| | **L = Max (Instance-loss, cluster-loss)** | *0.159* | *0.048* | *0.071* |
| | **L = Min (Instance-loss, cluster-loss)** | *0.306* | *0.401* | *0.194* |

Table 3: Exploring different loss functions beyond proposed method

