# OpenReview forum: "Reprodcibility of Contrastive Clustering"
_ML_Reproducibility_Challenge/2021/Fall — Reject_

### Official Review · Reviewer_fAtS · 2022-03-01
**RE:Reprodcibility of Contrastive Clustering**

**Rating:** 7
**Confidence:** 4

**Review:**

•	Reproducibility Summary: Provided

•	Scope of reproducibility: Yes
Reimplemented the results of pair-wise similarity analysis (Figure 3 of the paper), Clustering performance comparison (paper’s Table 2), and evolution of training process (Figure 4 of the paper) from the paper.

•	Code
They used authors code for Contrastive Clustering but created K-means method by themselves

•	Communication with original authors: No
There is no mention of communication with original authors

•	Hyperparameter search: No

•	Ablation Study: Yes
Ablation study is conducted to examine the importance of the contrastive heads


•	Discussion on results: Yes
They have discussed their results and also compared them with original paper’s results. They conclude that original paper’s results are credible.

•	Recommendations for reproducibility: No

•	Results beyond the paper: Yes
To measure the robustness of the proposed model, in addition to the original datasets, the authors applied the proposed method on other datasets including CIFAR-10, CIFAR-100, ImageNet-10

The authors also modified the loss function to improve the performance of the contrastive clustering method

•	Overall organization and clarity
Overall organization of the paper is good and easy to understand the experiments they have conducted and the discussion of the results. They clearly gave conclusions of their reproducibility experiments.

---

### Official Review · Reviewer_RHXw · 2022-04-04
**Not up to standard of scientific publications**

**Rating:** 2
**Confidence:** 4

**Review:**

This reproducibility report is a clear rejection in my opinion so I will take some of the review space to offer my opinion on how the authors can improve not only this report, but their research and any future work in general.

First off, let me start with something that should be rather easy to fix: language. The paper is not written in proper English, mostly notable by the abundance of incorrect capitalizations, missing determiners and generally unscientific language (e.g. "huge difference", Section 3.1.2). While the paper's quality greatly suffers from this, this is of course not the basis for my review. Thus, I am simply using this space to offer some suggestions to the authors on how to improve any future articles they may write. I suggest that the authors use tools such as Grammarly (there is a free tier available but the paid plan offers checking scientific language) in their future reports. A few spelling mistakes are not a problem and I would say even expected (I also make mistakes) but the incorrect use of language disturbs the reading flow of the paper.

Now, on to issues regarding the contents of the report.
- In Section 2, a sentence from the original paper has been copied without proper quotations ("Two separate MLPs, ...") while also leaving in descriptions of the figure they were copied from, and consequently being out of context.
- Generally the description of the original paper's method and experiments is insufficient in my eyes. Take for example the sentence "They also change the dataset and ...", where not further information is given about the actual datasets that have been used. This sentence leaves the reader wondering which dataset the original authors used.
- The figures and tables of the main results are put in the appendix which somehow disturbs the reading flow.
- There is actually a great difference between the reproduced Figure 2 and the original figure in that the negative sample similarity is much lower than the negative cluster similarity. This is unfortunately not further investigated. It could be the result of a different seed - just one additional run could have given you more information on this.
- While the reproducing authors ran the experiments for a much smaller amount of epochs (which is not a problem per se) they argue that the clustering performance can only increase with more epochs. While the original paper shows a better performance with more epochs, I wonder how they reproducing authors come to that conclusion as I do not see any reason to assume that no degenerate solutions could in principle be obtained when training for more epochs. It thus seems to be a baseless assumption.
- The authors state in Section 3.2 that something has been proven. NEVER use the word "proof" if you do not offer a mathematical proof. What you do is showing results that indicate/suggest which is very different.
- The ablation study in Section 3.3 is not actually an ablation study but rather a reproduction of an ablation study of the original paper. Under ablation study I understand experiments that were not yet presented in the original paper. An interesting ablation study would have been, for example, to test the robustness to slightly incorrect numbers of clusters, as those have to be specified a priori and are consequently an easy source of error.
- The authors provide an extension to the originally proposed loss function by augmenting it with either an average/max/min operation in Section 3.4. I do not understand the motivation behind the max and min operations as those are not differentiable. With max you could argue that you have some sort of alternation between which term is being differentiated through but when using the loss function with the min operation you only make the minimum smaller and never differentiate through the other term. Regarding using the average in the loss function: The authors do not seem to notice that this is equivalent to using the (originally proposed) sum of the individual loss terms and scaling the gradients. The same result could thus be achieved by using a smaller learning rate on the originally proposed loss function. This is not mentioned anywhere in the report which makes me suspect that it went unnoticed.

I have to report that the authors' names are mentioned in the contribution statement (Section 5) which undermines the double-blind peer review process. I have only seen this in the end after reading the whole report.

Finally, while a review like this can be crushing, I hope you don’t give up the hope but keep trying. Eventually you will get there (although likely not with this paper)!

---

### Meta-Review · Program_Chairs · 2022-04-07

**Recommendation:** Reject
**Confidence:** 5

**Metareview:**

While the reproducibility carried out is a great start, there are many issues (see Reviewer RHXw's comments) that need to be fixed before this work can be accepted.

---

### Decision · Program_Chairs · 2022-04-09

Reject